# Sexual-risk behaviours and HIV and syphilis prevalence among in- and out-of-school adolescent girls and young women in Uganda: A cross-sectional study

**Joseph K. B. Matovu**[1,2]*, **Justine N. Bukenya**[1], **Dickson Kasozi**[1], **Stephens Kisaka**[1], **Rose Kisa**[1], **Agnes Nyabigambo**[1], **Abdulaziz Tugume**[1], **John Baptist Bwanika**[1], **Levicatus Mugenyi**[3], **Irene Murungi**[3], **David Serwadda**[1], **Rhoda K. Wanyenze**[1]

**1** Makerere University School of Public Health, Kampala, Uganda, **2** Busitema University Faculty of Health Sciences, Mbale, Uganda, **3** The AIDS Support Organization, Kampala, Uganda

* jmatovu@musph.ac.ug

**Data Availability Statement:** All relevant data are within the manuscript and its Supporting information files.

## Abstract

### Background

Adolescent girls and young women (AGYW) are at increased risk of sexually transmitted infections (STIs). We assessed sexual-risk behaviours and HIV and syphilis prevalence among AGYW in Uganda to inform the design of target-specific risk-reduction interventions.

### Methods

This analysis utilizes data from 8,236 AGYW aged 10–24 years, collected in 20 districts, between July and August 2018. AGYW engaged in sexual-risk behaviour if they: a) reported a history of STIs; or b) had their sexual debut before age 15; or c) engaged in sex with 2+ partners in the past 12 months; or c) did not use or used condoms inconsistently with their most recent partners. We diagnosed HIV using *Determine*™ *HIV-1/2*, *Stat-Pak*™ *HIV-1/2* and *SD Bioline.* We used *SD Bioline* Syphilis test kits to diagnose syphilis and *Treponema Pallidum Hemagglutination Assay* for confirmatory syphilis testing. Comparison of proportions was done using Chi-square ($\chi^2$) tests. Data were analysed using STATA (version 14.1).

### Results

Of 4,488 AGYW (54.5%) that had ever had sex, 12.9% ($n = 581$) had their sexual debut before age 15; 19.1% ($n = 858$) reported a history of STIs. Of those that had ever had sex, 79.6% (n = 3,573) had sex in the 12 months preceding the survey; 75.6% ($n = 2,707$) with one (1) and 24.2% ($n = 866$) with 2+ partners. Condom use with the most recent sexual partner was low, with only 20.4% ($n = 728$) reporting consistent condom use while 79.6% ($n = 2,842$) reported inconsistent or no condom use. In-school AGYW were significantly less likely to have ever had sex (35.6% vs. 73.6%, $P<0.001$), to have had sexual debut before age 15 (7.7% vs. 15.5%, $P<0.001$) or to engage in sex with 2+ partners (5.3% vs. 15.8%,

**Funding:** This study was supported by a grant from the Global Fund through The AIDS Support Organization (Grant#: UGA-C-TASO-1449) awarded to Makerere University School of Public Health to conduct formative research on HIV, sexual and reproductive health and gender-based violence status among adolescent girls and young women in Uganda.

**Competing interests:** The authors have declared that no competing interests exist.

$P$<0.001). Consistent condom use was significantly higher among in-school than out-of-school AGYW (40.1% vs. 12.7%, $P$<0.001). Overall, 1.7% ($n$ = 143) had HIV while 1.3% ($n$ = 104) had syphilis. HIV and syphilis prevalence was higher among out-of-school than in-school AGYW (HIV: 2.6% vs. 0.9%; syphilis: 2.1% vs. 0.5%, respectively).

## Conclusion

In-school AGYW engaged in more protective sexual behaviors and had less HIV and syphilis than their out-of-school counterparts. These findings suggest a need for target-specific risk-reduction interventions stratified by schooling status.

## Introduction

Globally, adolescent girls and young women (AGYW) are still disproportionally affected by HIV. In 2019, AGYW (aged 15 to 24 years) in sub-Saharan Africa accounted for 24% of new HIV infections although they constituted 10% of the population [1]. Innovative HIV prevention interventions are urgently needed to stem the HIV tide in this population sub-group. One of these interventions is keeping girls in school [2–5]. In a 2015 study of the effect of increased primary schooling on adult women's HIV status in Malawi and Uganda, Behrman [2] found that a one-year increase in schooling decreased the probability of an adult woman testing positive for HIV by 0.03% in Uganda and 0.06% in Malawi. Rosenberg et al. [3] found significantly lower teenage pregnancy rates among in-school aged 12–18 years compared to out-of-school young women while De Neve et al. [5] found that in-school adolescents (aged 10–19 years) were about twice as likely as those out of school to report having abstained from sexual intercourse. School enrolment was also positively linked to increased HIV awareness and openness to discussing sexual and reproductive health (SRH) issues with parents, such as sexual partners and contraception, possibly reflecting increased demand for SRH knowledge [5]. Collectively, these studies suggest stark differences in sexual and reproductive health outcomes between in- and out-of-school adolescents.

The association between schooling status and HIV infection has been documented in previous studies. One study using Demographic and Health Survey data from nine DREAMS (Determined, Resilient, Empowered, AIDS-free, Mentored, and Safe) countries in eastern and southern Africa (Lesotho, Swaziland, Uganda, Kenya, Malawi, Mozambique, Tanzania, Zambia and Zimbabwe) found that being currently in school was associated with reduced odds of HIV infection among women aged 15–19 years in three of the nine countries (Lesotho, Swaziland and Uganda); however, there was no significant association between being in school and HIV infection in six of the nine countries (Kenya, Malawi, Mozambique, Tanzania, Zambia or Zimbabwe) [6]. However, another study conducted among young women aged 13–23 years in rural South Africa found that, over a period of 3.5 years of follow-up, the cumulative incidence of HIV was 19.9% among young women with low school attendance (<80% school days) versus 7.6% among young women with high school attendance (≥80% school days) [7]. The cumulative incidence for herpes simplex virus type 2 (HSV-2) followed a similar trend: 38.5% of young women with low school attendance had HSV-2 at the end of the follow-up period versus 17.3% among those with high school attendance. In addition, the weighted hazard of HIV and HSV-2 was greater for young women who attended less school than those who attended more school and among those who dropped out than those who stayed in school [7]. The findings from the latter study corroborate findings from previous studies conducted in South Africa and Zimbabwe which showed that out-of-school young women had three or

more times higher odds of HIV or HSV-2 infection than those who were in school [8–10]. These studies reaffirm the notion that keeping girls in school is crucial for improving their health outcomes, although further research is still needed to improve our understanding of the differences in risk-taking behaviors and the prevalence of HIV and other sexually transmitted infections (STI) between in- and out-of-school AGYW.

Thus, although available evidence is sufficient to confirm the association between schooling status and the risk of HIV infection [5–7, 9, 11], several studies did not include both behavioural and biomarkers in the same study while schooling status was defined using a self-reported question on highest level of education attained. Besides, some studies enrolled older adolescent girls (15–19 years) or young women (15–24 years) but did not include the very young adolescents aged 10–14 years. Besides, although previous studies assessed the prevalence of sexually transmitted infections (STIs) among AGYW, few studies include both HIV and syphilis in the same study, yet evidence shows that HIV and syphilis coinfection is common [12, 13]. This presents a missed opportunity for targeting the very young adolescents who are vulnerable to misinformation on sexual health matters and are at increased risk of HIV and other sexually transmitted infections (STIs). To bridge these apparent gaps, we assessed sexual-risk behaviours and HIV and syphilis prevalence among currently in-school and out-of-school AGYW aged 10–24 years to inform the design of appropriate STI prevention interventions for in- and out-of-school AGYW.

## Materials and methods

### Study site

The data used in this analysis were collected as part of large formative study to assess HIV, sexual and reproductive health and gender-based violence status among AGYW in Uganda. The large study was conducted in 233 villages and 80 schools in twenty (20) purposely selected districts of Uganda (Kalangala, Nakasongola, Kiboga, Buikwe, Jinja, Buyende, Kaliro, Bugiri, Tororo, Mbale, Bukwo, Busia, Hoima, Kyankwanzi, Kasese, Kisoro, Amolatar, Otuke, Amuru and Kitgum). These districts were selected from a list of forty priority districts that were targeted by The AIDS Support Organization (TASO) for the implementation of the Global Fund-supported Adolescent Girls and Young Women (AGYW) program in Uganda. TASO is one of the two Principal Recipients of the Global Fund grant in Uganda. The Uganda AGYW program is part of the Global Fund Strategy (2017–2022) to reduce new HIV infections among AGYW by 58% by 2022 in 13 sub-Saharan Africa countries including Botswana, Cameroon, Kenya, Lesotho, Malawi, Mozambique, Namibia, South Africa, Swaziland, Tanzania, Uganda, Zambia and Zimbabwe [13, 14]. The forty districts were selected for immediate targeting because they were located in regions with high HIV prevalence (above the national adult average of 6%) [15] and had high teenage and unwanted pregnancies [16]. The Uganda program targets in- and out-of-school AGYW aged 10–24 years and aims to reduce the number of new HIV infections among AGYW through social and behavior change communication (SBCC); vocational skilling; enterprise development assistance; and provision of second-education opportunities to out-of-school AGYW who are interested in undertaking non-formal, skills-based training in the target districts. The large study was conducted prior to implementation of the AGYW program and served to provide the baseline data needed to inform the design and implementation of the above-mentioned interventions.

### Study design and population

The large study was a cross-sectional, quantitative study conducted among in- and out-of-school AGYW aged 10–24 years, residing in 20 districts in Uganda. In-school AGYW were

those that were currently in school at the time of the survey while out-of-school AGYW were those who dropped out of school prior to school completion and had been out of school for at least one year prior to the survey. Thus, our criteria for enrolling out-of-school AGYW excluded AGYW who were not in school at the time of the survey because they completed school.

## Sample size determination

The sample size for the large study was determined using the formula for sample surveys suggested by Lwanga and Lemeshow [17]. Assuming a type-1 error of 5%, p1 = 0.336 and p2 = 0.616 (where 0.336 is the proportion of AGYW who left school and 0.616 is the proportion of AGYW who were still in school based on the 2014 Uganda National Population and Housing Census [18]), HIV-prevalence among AGYW aged 15–24 years is 3.3% [15], a margin of error of 0.05, and a non-response rate of 0.10 [19], we estimated that we would need to enroll 8,473 AGYW aged 10–24 years. Sample size was determined separately for each pre-specified district using an appropriate formula for sample surveys which accounts for the target population size (using census data) per district. For each district, the sample was proportionately distributed across the selected schools and villages depending on the size of the target population (using census data) in each school and village. This approach inherently accounted for the multi-level design effect. The number of AGYW to be interviewed in each district (stratified by age-group and schooling status) was calculated using the 2017 population estimates for each district, after adjusting the 2014 population size for a population growth rate of 3.0% per annum.

## Sampling procedures

Data for in-school AGYW were collected within the sampled schools while data for out-of-school AGYW were collected at the household level within the sampled villages. For this reason, sampling procedures were performed differently for schools and villages, respectively, as described below.

   **a) Sampling procedures for in-school respondents.**   In-school respondents were selected using multi-stage cluster sampling. In stage 1, a list of schools for each district was generated by the Study Biostatistician using a general master-list of schools obtained from the Ministry of Education and Sports (MoES) to serve as the sampling frame. In stage 2, four schools (1 primary school, 2 secondary schools, and 1 tertiary school, or otherwise, depending on the distribution of schools in each district) were randomly selected from each district using simple random sampling procedures, for a total of 80 schools in 20 districts. Within each district, schools were grouped according to school level—i.e. primary, secondary and tertiary—and unique numbers were assigned to each level. The assigned unique numbers were then written on pieces of paper that were placed in a box and churned thoroughly. An interviewer picked a piece of paper from the box without replacement. This process was repeated for each school level until all the four schools in each district had been selected. It is important to note that if there was only one school at a given level within a district, e.g. primary school, that school was immediately selected without any form of sampling. In stage 3, working with the assigned school teacher or head teacher, we obtained a list of pupils/students aged 10–24 years in each school from the school register. From this list, groups of pupils/students aged 10–14, 15–19 and 20–24 years were generated from which the number of pupils/students to be interviewed in each school was selected using systematic sampling procedures. The number of pupils/students to be selected per school was determined from a pre-determined district quota. Each district quota included an estimate of the number of AGYW to be interviewed per age category,

stratified by schooling status. Since participants in the age group 20–24 years who were still in school were difficult to find in the community, all in-school AGYW identified in each school were selected as part of the sample. In cases where we could not obtain the required sample in age-group 20–24 years from the selected schools, we substituted this age-group by interviewing AGYW aged 18–19 to obtain the district quota.

**b) Sampling procedures for out-of-school respondents.** The process of selecting out-of-school AGYW was done using multi-stage cluster sampling. In stage 1, a list of villages was generated for each study district by the Study Biostatistician using a general master-list of all census enumeration areas in Uganda, obtained from the Uganda Bureau of Statistics (UBOS). The census enumeration areas were generated for the Uganda National Population and Housing Census in 2014 [18] and have been in use since then. Using the UBOS enumeration areas, we randomly selected twelve (12) villages from each district list, for a total of 233 villages in all the 20 districts, using systematic random sampling procedures. The decision on surveying 12 villages per district was taken pragmatically while taking into consideration monetary, time (2 months), and logistical requirements needed to successfully execute the survey in 20 districts. In stage 2, a list of households with out-of-school AGYW aged 10–24 years within each village was generated in consultation with the village (Local Council I) chairperson. Identified households were then categorized into three strata representing households with adolescent girls and young women aged 10–14, 15–19 and 20–24 years. From each stratum, a list of households equivalent to the number of AGYW that had to be surveyed for each age-group was selected using systematic random sampling techniques, based on a pre-determined age-group-based distribution of out-of-school AGYW in each village. Using these procedures, we eventually selected 18 households per village for a total of 216 households in each district. In stage 3, we selected one AGYW per household (while ensuring adequate representation of the different age-groups) using simple random sampling techniques. If there was one eligible AGYW in a given household, that AGYW was immediately selected and invited to participate in the study. If the household had more than one eligible AGYW, we generated a list of names of all the eligible AGYW in the household from whom one AGYW was selected using simple random sampling procedures.

## Data collection procedures

Data were collected by ten field teams (each team was responsible for two districts) between July and August 2018 using paper-based questionnaires. Teams were deployed for fieldwork simultaneously. Field teams were trained in interviewing skills (including how to administer very sensitive questions (such as those on sexual behavior) to AGYW), questionnaire content and flow of questions, how to obtain informed consent from the participants, and (specific to laboratory and counselling personnel), standard operating procedures for sample collection and reporting; HIV and syphilis testing procedures, and provision of pre- and post-test counseling support to the respondents. The field teams were trained for three days ahead of field data collection. Questionnaires were translated into up to eight different languages in line with the languages spoken in the targeted districts. To improve clarity of questions, particularly to the very young adolescents, questions were phrased with illustrative examples, e.g. *Have you ever had any sexual intercourse in your life? (By this, I mean when a man or boy puts his penis in a woman or girl's vagina)*, and the study tools were pilot-tested in a non-study community in Kampala, Uganda, with 50 AGYW identified with the support of the village health teams. The feedback from the pilot-testing of tools helped the study team to improve clarity of translations and definition of unfamiliar terms.

Each team had five interviewers (three females and two males), one laboratory technician and one Nurse counsellor. While in the field, we assigned the very young adolescent girls (10–14 years) and those aged 15–17 years to female interviewers while the much older adolescents (e.g. 18–19 year-olds and those aged 20–24 years) were assigned to male interviewers. Interviewers conducted between 6–7 interviews per day, with each interview lasting approximately one-and-a-half hours. We collected data on socio-demographic characteristics, sexual behavior and history of self-reported sexually transmitted infections. At the end of each field day, the team reviewed the day's work, including number of interviews conducted (per age-group and schooling status), number of blood samples collected (and corresponding tests done) and compared the number of interviews to the number of tests done. Team Leaders submitted weekly reports to the Study Coordinator, and the study implementation team met weekly to discuss team performance and resolve any field challenges reported from the teams accordingly.

## HIV and syphilis testing procedures

HIV testing was done following the Ministry of Health's HIV testing algorithm [20]. Specifically, we used *Determine*$^{TM}$ *HIV-1/2* rapid test as a screening test; if results were non-reactive, these were reported as HIV-negative. If the results were reactive; the individual was subjected to *Stat-Pak*$^{TM}$ *HIV-1/2* as a confirmatory HIV test. After confirmatory HIV testing, reactive results were reported as HIV-positive while non-reactive results were subjected to *SD Bioline HIV-1/2* rapid test as a tie-breaker test. Reactive *SD Bioline HIV-1/2* rapid test results were reported as inconclusive while non-reactive results were reported as HIV-negative. Individuals with inconclusive results were advised to seek repeat HIV testing at the existing health facilities within 14 days of the first inconclusive results. For quality control purposes, all HIV-positive samples and 5% of HIV-negative samples were sent to the Central Public Health Laboratories (CPHL) in Kampala. CPHL is the technical focal point for Laboratory Services within the Ministry of Health and provides stewardship for the National Health Laboratory Network in Uganda. Syphilis testing was done using *SD Bioline* Syphilis test kits and participants were notified of their results on the same day. Confirmatory testing was done through re-testing of all syphilis-positive and 5% of syphilis-negative individuals at the CPHL in Kampala. To detect active syphilis infection, rapid plasma reagin (RPR) titers were used. The RPR card test was used in dilutions of 1:8. For confirmatory syphilis testing, the *Treponema pallidum hemagglutination assay* was used. Both HIV-positive and syphilis-positive clients were referred for follow-up care at the nearest health facilities, as appropriate.

## Measurement of variables

The dependent variables were: a) sexual-risk behavior and b) prevalence of HIV and syphilis infections, assessed separately among in- and out-of-school AGYW. Adolescent girls and young women were deemed to have engaged in sexual-risk behavior if they: a) reported a history of sexually transmitted infections; or b) reported that they had their first-time sexual experience before the age of 15; or c) had sexual intercourse with multiple (2+) sexual partners in the past 12 months; or d) did not use a condom or used condoms inconsistently with their most recent sexual partner. No attempt was made to create one composite variable of sexual-risk behaviors because each behavior was considered to constitute a level of HIV/STI risk on its own. HIV and syphilis prevalence was determined as a percentage of those tested who tested positive for HIV or syphilis.

The independent variables included age-group (categorized as 10–14, 15–19, and 20–24 years), highest level of education attained at the time of the survey (in-school AGYW were asked about their current class of attendance), marital status (categorized as 'never married',

'in a relationship but not married', 'married or in union', and 'divorced/widowed/separated'), history of HIV testing (ever tested for HIV; tested for HIV in the past 12 months), alcohol use before sex, wealth tertile (categorized as 'low', 'middle' and 'high'), comprehensive knowledge of HIV (categorized as 'low', 'medium' and 'high') and vulnerability index (categorized as 'low', 'medium' and 'high'). A detailed description of how wealth tertile, comprehensive HIV knowledge and vulnerability index were measured is presented below.

**Wealth tertile.** Responses on household possessions were used to create an index representing a wealth proxy for the AGYW interviewed. The list of household assets probed for included whether or not the respondent owned a home or lived in a family home; ownership of a radio, television set, bicycle, motorcycle, cell phone, regular (landline) phone, computer, income-generating business, indoor bathroom, running water either inside the house or inside the compound of the house, electricity, car, generator and solar electricity. To construct the socio-economic status (SES)/wealth index, each household item was assigned a weight ascertained through principal component analysis. Then, the scores were standardized in relation to a standard normal distribution with a mean of zero and a standard deviation of one. For each individual, the scores on household possessions were then summed up, ranked and subdivided into wealth tertiles (low, middle and high), depending on their scores, with each tertile containing a third of the participants.

**Comprehensive knowledge of HIV.** Comprehensive HIV knowledge was defined based on the following variables: a) knowing that consistent use of condoms during sexual intercourse and having just one uninfected, faithful partner can reduce the risk of getting HIV; b) knowing that a healthy-looking person can have HIV, and c) rejecting the two most common misconceptions about HIV transmission or prevention [21], namely: a) belief that one can acquire HIV from mosquito bites and b) belief that one can acquire HIV by sharing food with an HIV-infected person. To construct this index, responses to the above questions were assigned one and zero for a positive and negative response, respectively, and a weight ascertained through principal components analysis. Then, the scores were standardized in relation to a standard normal distribution with a mean of zero and a standard deviation of one. For each individual, the scores on the questions were then summed up; ranked and sub-divided into three knowledge levels (low, medium and high), depending on their scores, with each level containing a third of the participants.

**Vulnerability measures.** Vulnerability was measured at the individual, household and community levels, following the steps outlined in the report entitled, *The Adolescent Girls Vulnerability Index*: *Guiding Strategic Investment in Uganda* [22]. At the individual level, those aged 10–14 years were considered vulnerable if they were at least two years behind grade for age or were not in school and/or not living with their parents. For those aged 15–19 years, one was considered to have individual level vulnerability if she has ever been married, or given birth or currently married, or did not attend secondary school, or engaged in high-risk sex (sex under the age of 15 or multiple/non-regular partners). At the household level, a girl (10–19 years) was considered vulnerable if she experienced any two of the following five conditions: no access to improved source of water, no access to improved sanitation, household head has no education, food insecurity (no access to food in a day), and non-family support (ever consulted others for social support other than a family member). At the community level, a girl was considered vulnerable if she lived in a community characterized by any one of the following: high rate of early marriage before the age of 18, high rate of illiteracy, increased prevalence of HIV, and low comprehensive knowledge of HIV. At each level, a score of 1 was given if a girl experienced these measures and 0 if otherwise. We then used principal component analysis on the scored data (0/1) to derive the vulnerability index. The vulnerability index

was divided into tertiles (low, medium and high) with the highest tertile representing the most vulnerable group.

## Data analysis

Descriptive statistics such as frequencies and proportions were computed to summarize the characteristics of the study participants stratified by schooling status. HIV and syphilis status was determined out of all AGYW tested for both infections and their prevalence presented in form of percentages, i.e. percentage of those tested for HIV and syphilis. We employed the "*svy*" option in STATA to account for the survey design when estimating the prevalence of HIV and syphilis at district level. However, since our study is not powered to allow for the computation of weighted estimates at the school level, we only present unweighted HIV and syphilis prevalence estimates, stratified by schooling status. Chi-square ($\chi^2$) tests were performed to compare proportions across different categories. The results are presented in tables as appropriate. Data analysis was conducted using STATA statistical software (version 14.1).

## Ethical considerations

Ethics approval was provided by the Makerere University School of Public Health's institutional review board (Protocol#: 593) and the study protocol was cleared by the Uganda National Council for Science & Technology (Protocol#: SS 4678), as per national research regulations. Written informed consent was solicited from all respondents by a trained study team member prior to data collection. Willing participants signed two copies of the informed consent form, one for themselves and the other to be retained by the study team. Given the nature of the population targeted (10–24 years); informed consent was sought in three ways: a) for adolescent girls aged 10–17 years (who were not yet emancipated), we sought written parental consent for their daughters to participate in the study. If parental consent was granted, we sought written assent from the adolescents prior to enrolling them in the study; b) for adolescents aged 18–24 years—who are legally eligible to provide their own consent to participate in the study—written informed consent was obtained from them directly, and c) for adolescents aged 10–17 years who were "emancipated minors" (defined as those who were living on their own, or married), written informed consent was obtained from them directly without seeking parental consent first. The written informed consents provided to the study participants had detailed information about the study, including the risks and benefits, and emphasis on the protection of confidentiality. If, however, cases of intimate partner violence (IPV) were reported, the affected AGYW were referred to the nearest health facilities to receive appropriate support and management of the consequences of IPV.

## Results

### Respondents' characteristics

Table 1 shows the characteristics of the 8,236 respondents (97.2% of the total sample) who were enrolled into the large study, stratified by schooling status. Of these, 50.2% (n = 4,139) were in-school while 49.7% (n = 4,097) were out-of-school AGYW. A majority of the AGYW were aged 15–19 years (44.2%, n = 3,644) and 20–24 years (40%, n = 3,295) and had primary (40.9%, n = 3,369) or secondary education (41.6%, n = 3,429) as their highest level of education. Slightly more than one-third (36%, n = 2,966) described themselves as Catholics and 30.4% (n = 2,506) as Protestants. Sixty-two per cent (n = 2,530) of out-of-school AGYW were not able to read in their local language—a proxy measure of literacy. Sixty-four per cent (n = 5,247) had ever tested for HIV and received their HIV test results. A higher proportion of

**Table 1. Background characteristics of AGYW by schooling status.**

| Characteristic | Total N = 8,236 (%) | Schooling Status | |
| --- | --- | --- | --- |
| | | In-school N = 4,139 (%) | Out-of-School N = 4,097 (%) |
| **Overall** | **8,236 (100)** | **4,139 (100)** | **4,097 (100)** |
| **Age-group (years)** | | | |
| 10–14 | 1297 (15.8) | 987 (23.9) | 310 (7.8) |
| 15–19 | 3644 (44.2) | 1882 (45.5) | 1762 (43.0) |
| 20–24 | 3295 (40.0) | 1270 (30.7) | 2025 (49.4) |
| **Education[a]** | | | |
| None | 139 (1.7) | 0 (0.0) | 139 (3.4) |
| Primary | 3369 (40.9) | 820 (19.8) | 2549 (62.2) |
| Secondary | 3429 (41.6) | 2166 (52.3) | 1263 (30.8) |
| More than secondary | 1168 (14.2) | 1153 (27.9) | 15 (0.4) |
| Missing | 131 (1.6) | 0 (0.0) | 131 (3.2) |
| **Religion** | | | |
| Catholic | 2966 (36.0) | 1492 (36.0) | 1474 (36.0) |
| Anglican / Protestant | 2506 (30.4) | 1256 (30.3) | 1250 (30.5) |
| Moslem | 879 (10.7) | 338 (8.2) | 541 (13.2) |
| Pentecostal / Born Again / Evangelical | 1565 (19.0) | 862 (20.8) | 703 (17.2) |
| Other Religions | 320 (3.9) | 191 (4.6) | 129 (3.1) |
| **Marital status** | | | |
| Never married | 5001 (60.7) | 3328 (80.4) | 1673 (40.8) |
| In relationship but not married | 1535 (18.6) | 757 (18.3) | 778 (19.0) |
| Married/in union | 1318 (16.0) | 21 (0.5) | 1297 (31.7) |
| Divorced/Separated/Widowed | 382 (4.6) | 33 (0.8) | 349 (8.5) |
| **Literacy level[b]** | | | |
| Can't read at all | 3092 (37.5) | 562 (15.6) | 2530 (61.7) |
| Can read but with difficulty | 2575 (31.3) | 1501 (36.3) | 1074 (26.2) |
| Can read with ease | 2569 (31.2) | 2076 (50.2) | 493 (12.0) |
| **Ever tested for HIV** | | | |
| No | 2989 (36.3) | 1799 (43.5) | 1190 (29.0) |
| Yes | 5247 (63.7) | 2340 (56.5) | 2907 (71.0) |
| **HIV test in last 12 months** | | | |
| No | 1506 (28.7) | 857 (36.6) | 649 (22.3) |
| Yes | 3741 (71.3) | 1483 (63.4) | 2258 (77.7) |
| **Comprehensive knowledge of HIV** | | | |
| Low | 2074 (25.2) | 1031 (24.9) | 1043 (25.5) |
| Medium | 2340 (28.4) | 1136 (27.4) | 1204 (29.4) |
| High | 3822 (46.4) | 1972 (47.6) | 1850 (45.2) |
| **Wealth tertile** | | | |
| Low | 2754 (33.4) | 721 (17.4) | 2033 (49.6) |
| Middle | 2737 (33.2) | 1320 (31.9) | 1417 (34.6) |
| High | 2745 (33.3) | 2098 (50.7) | 647 (15.8) |
| **Vulnerability** | | | |
| Low | 2754 (33.4) | 2709 (65.5) | 45 (1.1) |
| Medium | 2737 (33.2) | 1373 (33.2) | 1364 (33.3) |
| High | 2745 (33.3) | 57 (1.4) | 2688 (65.6) |

[a]Education categories refer to the highest level of education attended, whether or not that level was completed.

[b]Assessed by asking respondents to read prepared text in their own local language as a proxy measure of literacy.

out-of-school AGYW reported that they had ever tested for HIV or tested for HIV in the 12 months preceding the survey (ever tested: 77.7%, n = 2,258; tested in the past 12 months: 71%, n = 2,907) than their in-school counterparts (ever-tested: 56.5%, n = 2,340; tested in the past 12 months: 63.4%, n = 1,483). Comprehensive knowledge of HIV was generally low (46.4%, n = 3,822), with no observed difference between in-school and out-of-school AGYW (47.6%, n = 1,972 vs. 45.2%, n = 1,850). Only one-third of AGYW (33.3%, n = 2,745) was in the highest wealth tertile with 50.7% (n = 2098) of in-school versus 15.8% (n = 647) of out-of-school AGYW being in the highest wealth tertile. Out-of-school AGYW were significantly more likely to be in the lowest wealth tertile than their in-school counterparts (49.6% vs. 17.4%, n = 721; $P<0.001$). With regard to vulnerability, a majority of the in-school AGYW were categorized as having a low level of vulnerability (65.5%, n = 2,709) while a majority of the out-of-school AGYW (65.6%, n = 2,688) were categorized as having a high level of vulnerability.

## Sexual debut experiences among AGYW

Table 2 shows the distribution of sexual debut experiences among AGYW, stratified by schooling status. Overall, 54.5% (n = 4,488) had ever had sex, with out-of-school AGYW significantly more likely to report that they had ever had sex than their in-school counterparts (73.6%, n = 3,014 vs. 35.6%, n = 1,474; $P<0.001$). Overall, 12.9% (n = 581) of the AGYW that had ever had sex reported that they had their sexual debut before the age of 15; this proportion was significantly higher among out-of-school than in-school AGYW (15.5%, n = 467 vs. 7.7%, n = 114; $P<0.001$). However, the proportion of those initiating sex between the ages of 15 and 17 was about four times higher than the proportion that initiated sex before age 15. For instance, while the proportion of in-school AGYW that initiated sex between ages 10–14 was 7.7% (n = 114), this proportion rose to 48.9% (n = 721) between ages 15–17. Among out-of-school AGYW, the proportion of AGYW initiating sex between the ages of 10–14 years was 15.5% (n = 467) but this increased to 51.2% (n = 1544) among those initiating sex between ages 15 and 17. In general, up to 63.4% of AGYW initiated sexual intercourse before age 18.

When asked with whom they had their sexual debut, a majority (85.1%, n = 3,821) of AGYW that had ever had sex reported that they had their first-time sex with a boyfriend, and this was true for both in- and out-of-school AGYW. However, out-of-school AGYW were significantly more likely to report that their first-time sexual partner was their husband than their in-school counterparts (15.6%, n = 471 vs. 0.4%, n = 6; $P<0.001$). Although only a small percentage of AGYW that had ever had sex (2.9%, n = 129) reported that they had their first-time sex with a close relative, teacher or another close person; it is important to note that this proportion was significantly higher among in-school than out-of-school AGYW (4.1%, n = 60 vs. 2.3%, n = 69; $P<0.001$).

With regard to age-disparity between AGYW and their first sexual partner, a higher proportion of in-school compared to out-of-school AGYW that had ever had sex (73%, n = 1,075 vs. 61.2%, n = 1,845) engaged in sex with male partners who were 1–4 years older than them. However, out-of-school AGYW were significantly more likely to report that they had their first-time sex with someone who was 5+ years older than them than their in-school counterparts (21.9%, n = 659 vs. 13.3%, n = 196; $P<0.001$). Nearly eighty-six per cent (n = 3,848) of the AGYW that had ever had sex reported that they were willing or somewhat willing to have sex at their sexual debut; with comparable proportions of out-of-school and in-school AGYW (84.4%, n = 1,245 vs. 86.3%, n = 2,603). Nearly half of the AGYW that had ever had sex (48.6%, n = 2,183) reported that they did something to protect themselves against pregnancy at first-time sex, with a significantly higher proportion of in-school

**Table 2. Sexual debut experiences of AGYW by schooling status.**

| Characteristic | Total N = 8,236 (%) | Schooling Status | | Chi Square |
| | | In-school N = 4,139 (%) | Out-of-School N = 4,097 (%) | P-value |
| --- | --- | --- | --- | --- |
| Overall | 8,236 (100) | 4,139 (100) | 4,097 (100) | |
| **Ever had sex** | | | | |
| No | 3748 (45.5) | 2665 (64.4) | 1083 (26.4) | <0.001 |
| Yes | 4488 (54.5) | 1474 (35.6) | 3014 (73.6) | |
| **Age at first sex[a]** | | | | |
| Before age 15 | 581 (12.9) | 114 (7.7) | 467 (15.5) | <0.001 |
| 15–17 years | 2265 (50.5) | 721 (48.9) | 1544 (51.2) | |
| 18+ years | 1597 (35.6) | 620 (42.1) | 977 (32.4) | |
| Age at first sex missing[b] | 45 (1.0) | 19 (1.3) | 26 (0.9) | |
| **With whom did you have your first-time sexual debut with?[a]** | N = 4488 | N = 1474 | N = 3014 | |
| Boyfriend | 3821 (85.1) | 1388 (94.2) | 2433 (80.7) | <0.001 |
| Husband | 477 (10.6) | 6 (0.4) | 471 (15.6) | |
| Close relative (father, brother, uncle, etc.) | 26 (0.6) | 13 (0.9) | 13 (0.4) | |
| Teacher or other close person | 103 (2.3) | 47 (3.2) | 56 (1.9) | |
| Others | 61 (1.4) | 20 (1.4) | 41 (1.4) | |
| **Age of person first had sex with[a]** | | | | |
| Same age | 334 (7.4) | 109 (7.4) | 225 (7.5) | <0.001 |
| Younger | 85 (1.9) | 21 (1.4) | 64 (2.1) | |
| 1–2 years older | 1647 (36.7) | 648 (44.0) | 999 (33.1) | |
| 3–4 years older | 1273 (28.4) | 427 (29.0) | 846 (28.1) | |
| 5+ years older | 855 (19.1) | 196 (13.3) | 659 (21.9) | |
| Don't know/remember | 294 (6.6) | 73 (5.0) | 221 (7.3) | |
| **Willingness to have sex at first-time sexual debut[a]** | | | | |
| Very willing | 3215 (71.6) | 966 (65.5) | 2249 (74.6) | <0.001 |
| Somewhat willing | 633 (14.1) | 279 (18.9) | 354 (11.7) | |
| Not willing at all | 590 (13.1) | 201 (13.6) | 389 (12.9) | |
| Don't know | 50 (1.1) | 28 (1.9) | 22 (0.7) | |
| **Pregnancy prevention at first sex[a]** | | | | |
| No | 2305 (51.4) | 525 (35.6) | 1780 (59.1) | <0.001 |
| Yes | 2183 (48.6) | 949 (64.4) | 1234 (40.9) | |
| **What did you use to prevent pregnancy at first sex?[a]** | | | | |
| Used a condom | 1902 (88.4) | 813 (86.3) | 1089 (90.1) | <0.001 |
| Other modern methods[c] | 150 (7.0) | 65 (6.9) | 85 (7.0) | |
| Traditional methods[d] | 99 (4.6) | 64 (6.8) | 35 (2.9) | |
| **Alcohol use at first sex[a]** | | | | |
| No | 4300 (95.8) | 1406 (95.4) | 2894 (96.0) | 0.321 |
| Yes | 188 (4.2) | 68 (4.6) | 120 (4.0) | |

[a]Expressed among those that reported that they had ever had sex.

[b]"Missing" represents AGYW who had ever had sex for whom age at first sex was not recorded.

[c]These methods include injectables, pills, rhythm method, emergency contraceptive pills, and implants.

[d]These methods include lactational amenorrhea method, withdrawal method and other methods.

AGYW reporting that they did so than their out-of-school counterparts (64.4%, n = 949 vs. 40.9%, n = 1,234; *P*<0.001). Although the numbers were pretty small, we found that alcohol use at first-time sex was slightly higher among in-school (4.6%, n = 68) than out-of-school AGYW (4%, n = 120).

## Number of sexual partners, condom use with most recent sexual partner and STI treatment-seeking behaviors

Table 3 shows the distribution of the different sexual-risk behaviors reported by AGYW that had ever had sex, stratified by schooling status. Of the 4,488 AGYW that had ever had sex, 3,573 (79.6%) reported that they had sex in the past 12 months. Of these, 75.6% (n = 2,707) reported that they had sex with one sexual partner while 24.2% (n = 866) reported that they engaged in sex with 2+ sexual partners. Out-of-school AGYW were significantly more likely to report that they engaged in sex with 2+ sexual partners in the past 12 months than their

**Table 3. Sexual-risk behaviors of AGYW that have ever had sex, stratified by schooling status.**

| Characteristic | Total (N, %) | Schooling Status | | P-value |
| | | In-school AGYW (n, %) | Out-of-School AGYW (n, %) | |
|---|---|---|---|---|
| **Had sex in the last 12 months (Yes)[a]** | 3,573 (79.6) | 1,008 (68.4) | 2,565 (85.1) | |
| **Number of sexual partners (Last 12 months)[b]** | | | | |
| 1 Partner | 2707 (75.6) | 789 (78.3) | 1918 (74.8) | 0.028 |
| 2+ Partners | 866 (24.2) | 219 (21.7) | 647 (25.2) | |
| **Most recent sexual partner (Last 12 months)[b]** | | | | |
| Boyfriend | 2215 (62.0) | 958 (95.0) | 1187 (46.3) | <0.001 |
| Husband | 1250 (35.0) | 19 (1.9) | 1308 (51.0) | |
| Other | 108 (3.0) | 31 (3.1) | 70 (2.7) | |
| **Condom use with most recent partner (Last 12 months)** | | | | |
| Always | 728 (20.4) | 404 (40.1) | 324 (12.6) | <0.001 |
| Sometimes | 785 (22.0) | 245 (24.3) | 540 (21.0) | |
| Rarely | 286 (8.0) | 85 (8.4) | 201 (7.8) | |
| Never | 1774 (49.6) | 274 (27.2) | 1500 (58.5) | |
| **Ever had a sexually transmitted infection (STI)[c]** | N = 4,488 | N = 1,474 | N = 3,014 | |
| No | 3630 (80.9) | 1143 (77.5) | 2487 (82.5) | <0.001 |
| Yes | 858 (19.1) | 331 (22.5) | 527 (17.5) | |
| **Sought STI treatment[d]** | N = 858 | N = 331 | N = 527 | |
| No | 167 (19.5) | 70 (21.2) | 97 (18.4) | 0.369 |
| Yes | 691 (80.5) | 261 (78.9) | 430 (81.6) | |
| **Time to STI treatment[e]** | N = 691 | N = 261 | N = 430 | |
| Same day | 59 (8.5) | 24 (9.2) | 35 (8.1) | 0.048 |
| Within 48hrs | 106 (15.3) | 52 (19.9) | 54 (12.6) | |
| Within a week | 291 (42.1) | 106 (40.6) | 185 (43.0) | |
| After 1 week | 235 (34.0) | 79 (30.3) | 156 (36.3) | |
| **Where sought STI treatment[e]** | | | | |
| Shop | 37 (4.8) | 9 (3.0) | 28 (5.9) | 0.195 |
| Pharmacy | 76 (9.9) | 34 (11.5) | 42 (8.8) | |
| Government Health Facility | 417 (54.1) | 150 (50.7) | 267 (56.2) | |
| Private Health Facility | 170 (22) | 76 (25.7) | 94 (19.8) | |
| Herbal/Traditional Provider | 26 (3.4) | 11 (3.7) | 15 (3.2) | |
| Other | 45 (5.8) | 16 (5.4) | 29 (6.1) | |

[a]Among those that had ever had sex;

[b]Among AGYW that reported sexual intercourse in the past 12 months.

[c]Among those that had ever had sex.

[d]Among those that reported a history of STI.

[e]Among those that sought treatment for the STI.

in-school counterparts (25.2%, n = 647 vs. 21.7%, n = 219; $P<0.001$). A majority (62.0%, n = 2,215) of those that had sex in the past 12 months reported that their boyfriend was their most recent sexual partner; 35% (n = 1,250) reported that their most recent partner was their husband, while 3.0% (n = 108) reported other categories of partners. In-school AGYW were significantly more likely to report that their most recent sexual partner was their boyfriend than their out-of-school counterparts (95.0%, n = 958 vs. 46.3%, n = 1187; $P<0.001$). However, out-of-school AGYW were significantly more likely to report that their most recent sexual partner was their husband than their in-school counterparts (51.0%, n = 1308 vs. 1.9%, n = 19; $P<0.001$).

When asked if they used a condom with their most recent sexual partner, only 20.4% (n = 728) of AGYW that had sex in the past 12 months reported consistent condom use (i.e. used a condom during all sexual encounters); 30.0% (n = 1,071) reported inconsistent condom use (i.e., used a condom sometimes or rarely), while 49.6% (n = 1,768) reported that they did not use a condom. A significantly higher proportion of in-school AGYW reported that they used condoms consistently (40.1%, n = 404 vs. 12.6%, n = 324; $P<0.001$) or that they used them sometimes or rarely with their most recent sexual partners than their out-of-school counterparts (32.7%, n = 330 vs. 27.3%, n = 701; $P<0.001$). However, out-of-school AGYW were significantly more likely to report that they *never* used condoms with their most recent sexual partner than their in-school counterparts (58.5%, n = 1500 vs. 27.2%, n = 274; $P<0.001$).

Nineteen per cent (n = 858) of AGYW that had ever had sex reported a history of sexually transmitted infections, with a higher proportion of in-school AGYW reporting that they had ever had a STI than their out-of-school counterparts (22.5%, n = 331 vs. 17.5%, n = 527; $P<0.001$). There was no significant difference in the proportion of in- and out-of-school AGYW who sought treatment for STI (78.9%, n = 261 vs. 81.6%, n = 430; $P = 0.369$). However, of those that sought treatment, out-of-school AGYW were significantly more likely to report that they delayed to seek treatment (i.e. sought treatment after 1 week of detection of signs and symptoms) than their in-school counterparts (36.3%, n = 156 vs. 30.3%, n = 79; $P = 0.048$). A majority of those that sought treatment reported that they sought treatment from government (54.1%, n = 417) and private health facilities (22%, n = 170), with no significant difference between in- and out-of-school AGYW.

## HIV and syphilis prevalence

Table 4 shows the prevalence of HIV and syphilis among AGYW that were enrolled in this study. Overall, 1.7% (n = 143) of the AGYW surveyed had HIV. HIV prevalence was significantly much higher among out-of-school than in-school AGYW (2.6%, n = 105 vs. 0.9%, n = 38; $P<0.001$). Across age-groups, HIV prevalence increased with increasing age from 0.6% (n = 8) among those aged 10–14 years, 1.1% (n = 40) among those aged 15–19 years to 2.9% (n = 95) among those aged 20–24 years. While HIV prevalence did not significantly differ between in- and out-of-school aged 10–14 and 15–19 years, HIV prevalence among 20–24 year-olds was significantly lower among those who were in school than those who were out of school (1.1%, n = 23 vs. 2.9%, n = 19; $P<0.001$).

HIV prevalence decreased with increasing wealth tertiles from 2.6% (n = 58) among AGYW in the lowest tertile; 1.6% (n = 43) among those in the middle tertile and 1.5% (n = 42) among those in the highest tertile. However, even then, HIV prevalence differed between in- and out-of-school AGYW within the same tertile, with in-school AGYW significantly more likely to have lower HIV prevalence than their out-of-school counterparts. For instance, among those in the lowest tertile, HIV prevalence was significantly higher among in-school

**Table 4. Distribution of HIV and syphilis prevalence by schooling status and selected sexual-risk behaviors.**

| Characteristic | HIV Infection (Unweighted) | | | Syphilis Infection (Unweighted) | | |
|---|---|---|---|---|---|---|
| | Total n (%) | In-school n (%) | Out-of-School n (%) | Total n (%) | In-school n (%) | Out-of-School, n (%) |
| Overall | 143/8,236 (1.7) | 38/4,139 (0.9) | 105/4,097 (2.6) | 104/8,236 (1.3) | 19/4,139 (0.5) | 85/4,097(2.1) |
| **Age-group (years)** | | | | | | |
| 10–14 | 8 (0.6) | 5 (0.5) | 3 (1.0) | 7 (0.5) | 7 (0.7) | 0 (0.0) |
| 15–19 | 40 (1.1) | 16 (0.9) | 24 (1.3) | 33 (0.9) | 6 (0.3) | 27 (1.5) |
| 20–24 | 95 (2.9) | 17 (1.3) | 78 (3.9) | 64 (1.9) | 6 (0.5) | 58 (2.9) |
| **Wealth tertile[a]** | | | | | | |
| Low (-2.3, -1.0) | 58 (2.1) | 5 (0.7) | 53 (2.6) | 43 (1.6) | 4 (0.6) | 39 (1.9) |
| Middle (-1.0, 0.3) | 43 (1.6) | 10 (0.8) | 33 (2.3) | 38 (1.4) | 8 (0.6) | 30 (2.1) |
| High (0.3, 7.4) | 42 (1.5) | 23 (1.1) | 19 (2.9) | 23 (0.8) | 7 (0.3) | 16 (2.5) |
| **Vulnerability[a]** | | | | | | |
| Low (-2.1, -1.1) | 16 (0.6) | 16 (0.6) | 0 (0.0) | 10 (0.4) | 10 (0.4) | 0 (0.0) |
| Medium (-1.1, 0.6) | 28 (1.0) | 21 (1.5) | 7 (0.5) | 25 (0.9) | 9 (0.7) | 16 (1.2) |
| High (0.6, 6.9) | 99 (3.6) | 1 (1.8) | 98 (3.6) | 69 (2.5) | 0 (0.0) | 69 (2.6) |
| **Age at first sex** | | | | | | |
| Never | 27 (0.7) | 20 (0.8) | 7 (0.7) | 18 (0.5) | 12 (0.5) | 6 (0.6) |
| Below 15 | 15 (2.6) | 1 (0.9) | 14 (3.0) | 15 (2.6) | 1 (0.9) | 14 (3.0) |
| 15–17 years | 58 (2.6) | 9 (1.3) | 49 (3.2) | 46 (2.0) | 4 (0.6) | 42 (2.7) |
| 18+ years | 43 (2.7) | 8 (1.3) | 35 (3.6) | 25 (1.6) | 2 (0.3) | 23 (2.4) |
| **Condom use at first sex** | | | | | | |
| No | 69 (2.7) | 7 (1.1) | 62 (3.3) | 51 (2.0) | 3 (0.5) | 48 (2.5) |
| Yes | 47 (2.4) | 11 (1.3) | 36 (3.2) | 35 (1.8) | 4 (0.5) | 31 (2.8) |
| **Condom use with most recent sexual partner (Last 12 months)** | | | | | | |
| Always | 13 (1.8) | 6 (1.5) | 7 (2.2) | 9 (1.2) | 1 (0.2) | 8 (2.5) |
| Sometimes | 22 (2.8) | 2 (0.8) | 20 (3.7) | 22 (2.8) | 2 (0.8) | 20 (3.7) |
| Rarely | 12 (4.2) | 1 (1.2) | 11 (5.5) | 6 (2.1) | 0 (0.0) | 6 (3.0) |
| Never | 49 (2.8) | 5 (1.8) | 44 (2.9) | 35 (2.0) | 3 (1.1) | 32 (2.1) |
| **Number of sexual partners (Last 12 months)** | | | | | | |
| No Sex | 47 (1.0) | 24 (0.8) | 23 (1.5) | 32 (0.7) | 13 (0.4) | 19 (1.2) |
| 1 Partner | 64 (2.4) | 10 (1.3) | 54 (2.8) | 49 (1.8) | 3 (0.4) | 46 (2.4) |
| 2+ Partners | 32 (3.7) | 4 (1.8) | 28 (4.3) | 23 (2.7) | 3 (1.4) | 20 (3.1) |
| **Comprehensive knowledge of HIV[a]** | | | | | | |
| Low (-4.5, -0.8) | 30 (1.4) | 7 (0.7) | 23 (2.2) | 27 (1.3) | 8 (0.8) | 19 (1.8) |
| Medium (-0.8, 0.2) | 49 (2.1) | 9 (0.8) | 40 (3.3) | 30 (1.3) | 4 (0.4) | 26 (2.2) |
| High (0.2, 1.3) | 64 (1.7) | 22 (1.1) | 42 (2.3) | 47 (1.2) | 7 (0.4) | 40 (2.2) |

[a]Obtained using Principal Component Analysis (PCA).

than out-of-school AGYW (0.7%, n = 5 vs. 2.6%, n = 52; $P<0.001$) and this was the case among in- and out-of-school AGYW in the highest tertile (1.5%, n = 23 vs. 2.9%, n = 19; $P<0.001$). Besides, HIV prevalence increased with increasing levels of vulnerability, from 0.6% (n = 16) among those with low levels of vulnerability, 1.0% (n = 28) among those with medium levels of vulnerability to 3.6% (n = 99) among those with high levels of vulnerability. This observation was true for both in- and out-of-school AGYW. However, out-of-school AGYW with high levels of vulnerability were significantly more likely to have higher HIV prevalence

than in-school AGYW at the same level of vulnerability (3.6%, n = 98 vs. 1.8%, n = 1; $P<0.001$). HIV prevalence increased with increasing numbers of sexual partners in the past 12 months, from 1.0% (n = 47) among those who reported that they did not engage in sex during this period, 2.4% (n = 64) among those who reported engaging in sex with only one sexual partner in the past 12 months to 3.7% (n = 32) among those who reported engaging in sex with 2+ sexual partners during this period. This observation was true for both in- and out-of-school AGYW; however, out-of-school AGYW had much higher HIV prevalence at all levels than their in-school counterparts.

Syphilis prevalence followed a similar trend as that for HIV with much higher levels reported among out-of-school AGYW than among in-school AGYW across age-group, wealth quintile, levels of vulnerability and number of sexual partners in the past 12 months. Overall, 1.3% (n = 104) had syphilis; 0.5% (n = 19) among in-school and 2.1% (n = 85) among out-of-school AGYW.

## Discussion

Our analysis of sexual-risk behaviors and HIV and syphilis prevalence among in- and out-of-school AGYW shows that: a) in-school AGYW were significantly less likely to engage in sex at an early age, and when they eventually engaged in sex, they were more likely to engage in first-time protected sex than their out-of-school counterparts; b) out-of-school AGYW were significantly more likely to engage in riskier sexual behaviors with less protection, and c) HIV and syphilis prevalence were significantly much higher among out-of-school than among in-school AGYW. These findings are highly correlated with wealth tertile and vulnerability levels: in-school AGYW were more likely to be in the highest wealth tertile with low levels of vulnerability while out-of-school AGYW were more likely to be in the lowest wealth tertile with high levels of vulnerability. As confirmed in previous studies [23, 24] as well as in our study, high levels of vulnerability were associated with high HIV and syphilis prevalence levels while being in the highest wealth tertile was associated with low HIV and syphilis prevalence levels. These findings suggest a need for stratified STI prevention interventions for in- and out-of-school AGYW that take into consideration differentials in vulnerability and wealth index between the two groups.

Our finding that the prevalence of both HIV and syphilis was much higher among out-of-school than in-school AGYW is consistent with previous findings [4, 6, 7, 25, 26] but not surprising given that out-of-school AGYW were more likely to engage in sex with multiple partners, to be less likely to use condoms consistently with these partners, and to engage in age-disparate relationships than their in-school counterparts. These factors have also been associated with both incident and prevalent HIV infection in previous studies [27–30]. Study findings suggest a need for interventions to keep girls in school, since evidence shows that staying in school likely restricts the time that in-school AGYW have to get in touch with older men as sexual partners, thereby reducing their HIV infection risk [30]. These findings also call for integrated HIV prevention interventions, including integration of economic strengthening components into HIV prevention interventions, since these integrated interventions have been shown to reduce sexual-risk behaviors among out-of-school AGYW [31–33].

Interestingly, in-school AGYW were significantly less likely to report that they had ever engaged in sex (35.6% vs. 73.6%) and, among those that had ever had sex, in-school AGYW were significantly less likely to report that they had their sexual debut before age 15 (7.7% vs. 15.5%) than their out-of-school counterparts. We also found that in-school AGYW were more likely to report first-time protected sex than their out-of-school counterparts, suggesting a need to educate all AGYW, but most importantly out-of-school AGYW, about the need for

correct and consistent use of protection at any sexual encounter, including the first sexual encounter, to reduce the risk of HIV/STI infection and teenage/unwanted pregnancies. Our finding that a higher proportion of out-of-school AGYW engaged in sexual debut before the age of 15 than their in-school counterparts is consistent with findings from other studies in Uganda and elsewhere [15, 26, 34], which improves their wider generalizability. In particular, findings from the Uganda Population-based HIV Impact Assessment (UPHIA) show that the percentage of young Ugandan females (15–24 years) who had sex before the age of 15 decreased with increasing levels of education from 20.1% among those with no formal education, 10.7% among those who completed primary education and 2.1% among those that completed secondary education or higher [14]. In a synthesis of national representative Demographic and Health Surveys data from 33 countries in sub- Saharan Africa (covering the period between 2004 and 2015), Melesse et al. [34] found that girls with less education (none or primary) initiated sex 2.2 years earlier, were married 4.4 years earlier and had their first child 2.5 years earlier than girls with secondary or higher education. These results re-affirm the need for integrated multidimensional interventions (including conditional and unconditional cash transfers, savings-led economic empowerment schemes, among others) that can help to not only keep girls in school but also help to improve their health outcomes [4, 35, 36].

We found that the proportion of AGYW initiating sex between ages 15 and 17 was four times higher than the proportion initiating sex between 10–14 years, with a much higher proportion of out-of-school AGYW initiating sex between ages 15 and 17 than their in-school counterparts. This observation implies that by age 17, up to 63.4% of AGYW have had their sexual debut but the biggest proportion of those initiating sex will have had their first-time sexual experience after age 15. These findings suggest that interventions aimed at delaying sexual debut should target the very young age-group of 10–14 years before they become sexually active. These findings also call for a need to target those aged 15–17 years with correct information about safer sexual practices, including safer pregnancy prevention options, since girls are likely to receive a lot of misinformation about sex and reproductive health from their peers during this period [37]. Studies conducted elsewhere [38–40] have confirmed that parents, and especially the mother, can be a useful and trusted source of sexual health information for adolescent girls. Therefore, it may be crucial for adolescent health programs to target parents with the right information and skills-building sessions to improve their self-efficacy to provide correct sexual and reproductive health information to their young daughters.

We also found that in-school AGYW were significantly more likely to report that they had their sexual debut with male partners who were 1–4 years older than them than out-of-school AGYW who were more likely to report that they had their first-time sex with male partners who were 5+ years older than them. Engaging in age-disparate sexual relationships may decrease the girls' ability to negotiate safe sex and increase the risk of teenage and unwanted pregnancies, and the risk of getting infected with HIV and other sexually transmitted infections [28, 29]. Our findings are in direct consonance with findings from prior studies that show that young women who stay in school and who attend school more frequently have partners closer to their age and fewer partners than young women who attend less school or drop out [11, 25, 30]. Collectively, these findings suggest a need to target out-of-school AGYW with unique interventions that can reduce their vulnerability to HIV infection, including those that can help them to reduce the number of sexual partners they have and/or help to improve their efficacy to insist on protected sexual intercourse at all times.

Our study had some limitations and strengths. Similar to other observational studies, our study is liable to recall bias especially on questions that stretched as far back as 12 months from the time of the survey. We tried to minimize recall bias by asking questions that pertained to more recent events, e.g. condom use with their most recent (or current) sexual partner. It is

also likely that some AGYW did not feel comfortable responding to questions on sexual behavior, given the sensitivity of these questions, e.g. questions on age at sexual initiation and number of sexual partners in the past 12 months. The fact that some of the older adolescents and young women were interviewed by male interviewers could have also affected AGYW's responses to these questions. However, we assigned same-sex interviewers to the very young adolescents (10–14 years) and those aged 15–17 years, where appropriate, to improve their ability to respond to the interview questions. As a result, we did not record any cases of incomplete questionnaires arising from the fact that respondents had failed to respond to some questions, including the very sensitive questions.

It is important to note that while the data are clustered at multiple levels (district, schools, villages), we only accounted for clustering at the district level while estimating the sample size but not at the school or village level and this is likely to have affected the precision of our sample size estimation. Furthermore, our paper could have been strengthened if we performed regression analyses to identify the factors that are independently associated with HIV or syphilis infection or engagement in sexual-risk behaviors. However, we performed descriptive statistics and any comparisons made between groups were done using Chi-square tests. Our analysis was not informed by any hypothesis-driven questions which could have helped to guide the analysis as well as further strengthen the presentation of findings. Nevertheless, we believe that the findings presented in this paper can help to inform the design of target-specific HIV/STI prevention interventions for AGYW not only in Uganda but also in other countries where differing levels of HIV/STI risk still exist between in- and out-of-school AGYW aged 10–24 years.

The above-mentioned limitations notwithstanding, our study had several strengths. This study was conducted among 8,236 AGYW across 20 districts which provides a large sample to generate useful population-level estimates to inform programming. Also, the study included both observational data and biomarkers, which enabled us to assess if sexual-risk behaviors (e.g. self-reported number of sexual partners in the past 12 months) were linked to the observed levels of HIV and syphilis infection. Our decision to focus on both HIV and syphilis was informed by prior evidence that shows that coinfection with HIV and syphilis is common [12, 13]. Indeed, Lynn and Lightman have described HIV and syphilis co-infection as a "dangerous combination" [13]. Most importantly, our study included adolescents aged 10–14 years; a population sub-group that is often missed in most population-based studies. The inclusion of the very young adolescents has enabled us to document sexual-risk behaviors and HIV and syphilis prevalence among 10–14 year-olds to inform programming for this age-group. We present findings stratified by schooling status, making it possible to show differentials in sexual-risk behaviors and prevalence of HIV and syphilis by whether the AGYW was in- or out-of-school. This is crucial for the design of more target-specific interventions rather than designing interventions that are presumed to be appropriate to all AGYW, which is less effective. Finally, our study interviewed in-school AGYW at school and did not depend on self-reports of being in-school, making it possible to make accurate comparisons between these two groups in terms of sexual-risk behaviors and the prevalence of HIV and syphilis.

## Conclusion

Our study shows marked differences in sexual-risk behaviors and the prevalence of HIV and syphilis between in- and out-of-school AGYW. We found that: a) in-school AGYW were significantly less likely to engage in sex at an early age, and when they eventually did, they were more likely to engage in protected sex than their out-of-school counterparts; b) out-of-school AGYW were significantly more likely to engage in riskier sexual behaviors (e.g. 2+ sexual

partners in the past 12 months) with less protection, and c) HIV and syphilis prevalence were significantly much higher among out-of-school than among in-school AGYW. The observed high prevalence of HIV and syphilis among out-of-school AGYW could be related to their engagement in high sexual-risk behaviors and age-disparate sexual partnerships coupled with their high levels of vulnerability. These findings suggest a need for interventions that can help to keep girls in school; and among those that are already out of school, there is a need for unique interventions to reduce their risk-taking behaviors, improve their ability to negotiate for safer sex, and reduce their vulnerability to the risk of HIV and other sexually transmitted infections.

## Supporting information

**S1 Data. Dataset used during the analysis of data.**
(DTA)

**S1 Questionnaire. Study questionnaire.**
(PDF)

## Acknowledgments

We would like to acknowledge members of the field team (interviewers, counsellors, laboratory technicians and team leaders) who collected the data and conducted HIV and syphilis testing in the 20 districts surveyed; the adolescent girls and young women who participated in the study and staff in the study coordination unit, with special thanks to Ms. Noor Ssekimpi for her administrative support. We are grateful to the district administrators for permission to conduct the study in the respective study districts.

## Author Contributions

**Conceptualization:** Joseph K. B. Matovu, Justine N. Bukenya, Irene Murungi, Rhoda K. Wanyenze.

**Data curation:** Levicatus Mugenyi.

**Formal analysis:** Joseph K. B. Matovu, John Baptist Bwanika, Levicatus Mugenyi.

**Funding acquisition:** Joseph K. B. Matovu, Irene Murungi, David Serwadda, Rhoda K. Wanyenze.

**Investigation:** Justine N. Bukenya, Stephens Kisaka.

**Methodology:** Joseph K. B. Matovu, Justine N. Bukenya, Dickson Kasozi, Rose Kisa, Agnes Nyabigambo, Abdulaziz Tugume, Levicatus Mugenyi, Irene Murungi, David Serwadda, Rhoda K. Wanyenze.

**Project administration:** Dickson Kasozi.

**Supervision:** Joseph K. B. Matovu, Justine N. Bukenya, Dickson Kasozi, Stephens Kisaka, Rose Kisa, Agnes Nyabigambo, Abdulaziz Tugume, John Baptist Bwanika, Levicatus Mugenyi, David Serwadda, Rhoda K. Wanyenze.

**Writing – original draft:** Joseph K. B. Matovu, Justine N. Bukenya.

**Writing – review & editing:** Joseph K. B. Matovu, Justine N. Bukenya, Dickson Kasozi, Stephens Kisaka, Rose Kisa, Agnes Nyabigambo, Abdulaziz Tugume, John Baptist Bwanika, Levicatus Mugenyi, Irene Murungi, David Serwadda, Rhoda K. Wanyenze.

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
