## [Decision Letter · Decision Letter 0]

21 May 2021

PONE-D-21-03834

Sexual-risk behaviours and HIV and syphilis prevalence among in- and out-of-school adolescent girls and young women in Uganda: a cross-sectional study

PLOS ONE

Dear Dr. Matovu,

Thank you for submitting your manuscript to PLOS ONE. After careful consideration, we feel that it has merit but does not fully meet PLOS ONE’s publication criteria as it currently stands. Therefore, we invite you to submit a revised version of the manuscript that addresses the points raised during the review process. You are welcome to rebut comments, but please consider those that will help clarify the study's methods, findings, and context to your readers. Where there are limitations to the work done, please acknowledge these and state why the manuscript adds value to the evidence base in this area.

We look forward to receiving your revised manuscript.

Kind regards,

Susan Marie Graham, MD, MPH, PhD

Academic Editor

PLOS ONE

Journal Requirements:

[This study was supported by a grant from The AIDS Support Organization (TASO) to Makerere University 615School of Public Health to conduct formative research on HIV, sexual and reproductive health and gender-616based violence status among adolescent girls and young women in Uganda]

 [The funders had no role in study design, data collection and analysis, decision to publish, or preparation of the manuscript]

Reviewers' comments:

Reviewer's Responses to Questions

**Comments to the Author**

1. Is the manuscript technically sound, and do the data support the conclusions?

Reviewer #1: Yes

Reviewer #2: Yes

Reviewer #3: Partly

2. Has the statistical analysis been performed appropriately and rigorously? 

Reviewer #1: No

Reviewer #2: Yes

Reviewer #3: No

3. Have the authors made all data underlying the findings in their manuscript fully available?

Reviewer #1: Yes

Reviewer #2: Yes

Reviewer #3: Yes

4. Is the manuscript presented in an intelligible fashion and written in standard English?

Reviewer #1: Yes

Reviewer #2: Yes

Reviewer #3: Yes

5. Review Comments to the Author

Reviewer #1: Overall, the paper is well written. However, the authors examine a subject that is exhausted. It is not clear what is novel about this paper as the main findings from this paper have been well established in several papers as the authors rightly admit. Instead one might expect that interventions to keep AGYW in school and economic empowerment programs are a more critical topic and such interventions should be the subject of current inquiry. Also, there are some major data analytical flaws that need to be addressed.

Major comments

The authors present a standard definition of the AGYW as those aged 15 to 24 in the introduction. However, the authors enrolled participants from 10 years. Is it correct to refer to the study group with a wider age bracket as AGYW?

In line 120, the authors state that further research is needed to understand the reasons for the differences in the risk of HIV and other STIs between in and out of school AGYW. This would have been a good addition to the body of knowledge, however it was not explored in this paper. To answer this question, one might have expected a qualitative inquiry.

Line 132-137 is a description of the strength of the study and should be moved to the discussion section. It is very unusual to present the number of study participants enrolled in the introduction section.

In line 188, the authors state for a total of “80 schools in 20 districts”. Was this planned, and if so, it should be stated explicitly. Also placing this information in the brackets takes away its significance.

In line 220, the authors explain the sampling at household level, however, it is not clear how the representation was achieved without using a stratified approach in their sampling approach.

In line 241, how did the investigators determine which AGYW would be interviewed by men, and which ones by females?

The authors should provide examples of questions that the AGYW answered. It is not clear to the reader how these questions were structured especially to fit a young audience of 10 years, and if they really understood these questions.

Ethical-legal issues arise from cases where minors report sexual abuse and it is not clear what was done. A statement on this issue is important.

Definition of study outcome: The sexual risk behavior study outcome appear to be several. In the analysis, all these items have remained separate. The authors did not attempt to create a composite index which would bring all these together.

Sample size calculation: The data are clustered at multilevels namely district, schools or villages. This was not taken into account at the sample size calculation to adjust for the potential design effect.

Data analysis: The authors have presented only crude results which could potentially be affected by confounding. One would expect the measured of effect would be attenuated if an adjusted analysis were conducted.

In the same line with adjusting for confounding, the authors did not adjust for the clustering effect. This may be accomplished using the Generalized estimating equation (GEE) models

Also, the authors did not take into account the survey design especially if they used the enumeration areas. The “svy” option in STATA would help to offset that.

The % of HIV positives in-school were lower than those out-of-school. This may be explained by the higher risk sexual behaviors. However, is it also possible that HIV positive girls drop out of school when they learn about their HIV status. It is not clear here what the chicken and what is the egg. However, the authors seem to assert that girls drop out of school and then they become HIV infected.

The results within Table 2 and 3 have varying denominators depending on whether an AGYW has had sexual intercourse before. To avoid confusion, the authors should revise the table and include the (n) against the variable to clarify on how many answered this question since the denominator at the top of the table of schooling status is not applicable. For example the question age at first sex is only answered by those who had ever had sex, yet based on the table it appears as if the denominator is all the 8236, but is actually half of that.

The mode of presentation of results needs to take into consideration the denominator even in the text. For instance in line 417 to 419 the authors state that “Nearly eighty-six per cent (n=3,848) of the AGYW reported that they were willing or somewhat willing to have sex at their sexual debut; with comparable proportions of out-of school and in-school AGYW (84.4%, n=1,245 vs. 86.3%, n=2,603)” There is a need to clarify that this is ‘among those who had ever had sex’. The same applies to line 431, and the authors should clarify this throughout the manuscript.

For table 4, some results appear in the text but not in the table. The RR’s should be presented in the table as well and not just the text. Also, please explain the rationale for RRs in the data analysis section, given this is a cross sectional study. What form of regression was used to generate these results and explain the choice. Why did the authors not conduct a multivariable regression analysis?

As already mentioned, no multivariable results are presented. For instance, is wealth tertile independently associated with HIV infection regardless of the schooling status? The same would apply to the sexual risk behaviors.

Overall, the data analysis lacks sufficient rigor and needs to be re-examined extensively.

The argument in line 554-556 is inadequately presented and could as well be removed.

Minor comments

The second sentence (line 57/58) in the results in abstract section specifically refers to the out of school adolescents who are not able to read. It is not clear why the authors specifically focus on this subgroup. The reader might expect overall rate of literacy or present the two subgroups for comparison.

Reviewer #2: Sexual-risk behaviours and HIV and syphilis prevalence among in- and out-of-school

adolescent girls and young women in Uganda: a cross-sectional study

General comments:

Thank you for the opportunity to review this piece of work. Overall, the paper is well-written and presents an important public health problem. However, it is very descriptive and could have been strengthened by a theoretical framework or be hypothesis-driven question and simple regression analysis to control for confounders.

Specific comments:

1. Abstract – line 57 – it is not clear what the sample size is and how many AGYW were recruited in the study across the 20 districts? What is the expected age-range for in-school as only 50% in school sounds very low? Please clarify

2. Abstract line 57/58 – was not being able to read text in local language a measure of literacy?

3. Line 165 – definition of out-of-school is it only based on duration out of school? How do you account for those who had completed the highest level and hence did not need to be in school? Maybe good to distinguish between out of school out of employment and completed school with dropouts. Adding an age factor may assist in defining who is a school drop out and hence more vulnerable maybe, and also identify repeat graders – those staying in school beyond the age-grade level. Do you have data on those who were still in school rather than ‘highest level education reached’?

4. Line 245 editing questionnaires or maybe quality control? Please clarify

5. Line 366 – I am not sure what we are measuring with ability to read in local language – is it literacy – if so please indicate? What was done with those who couldn’t read/write when obtaining consent?

6. Line 432 – do we know the relationship with the most recent partner as it determines whether condoms are used or not and to what extent they are used?

7. I have a problem with comparing HIV and STI prevalence (and any key outcomes) between in and out of school without controlling for key confounders such as age, important sexual behaviors that predispose AGYW to HIV infection as well structural factors that make AGYW vulnerable. A simple multivariable analysis would have taken the analysis and interpretations a step further.

8. In discussion – the authors refer to first-time protected sex, does this have any implication in terms of risk of acquiring HIV or syphilis? I think the message to drive is the need for correct and consistent use of protection and differentiated care for AGYW in and out of school.

9. Line 536-540 since this study was cross-sectional it is important to be cautious of making assumption that HIV prevalence was high due to factors that were not measured in time in the study.

Reviewer #3: My comments have been attached. .

6. PLOS authors have the option to publish the peer review history of their article (what does this mean?). If published, this will include your full peer review and any attached files.

Reviewer #1: No

Reviewer #2: No

Reviewer #3: **Yes: **Mayibongwe L. Mzingwane (PhD)

---

## [Author Response · Author response to Decision Letter 0]

21 Jun 2021

June 15th, 2021

The Editor 

PLoS ONE

Dear Sir,

Re: RESPONSE TO COMMENTS RAISED ON MS#: PONE-D-21-03834

Please find enclosed our revised manuscript based on the comments from the peer-reviewers.

We are glad for the opportunity to revise the paper, which has improved clarity of the main message in the paper. We than the reviewers for their insightfulness. We look forward to our paper being published in your prestigious journal.

Funding information:

This study was supported by a grant from The AIDS Support Organization (TASO) to Makerere University School of Public Health to conduct formative research on HIV, sexual and reproductive health and gender-based violence status among adolescent girls and young women in Uganda.

Regards,

Joseph KB Matovu, MHS, PhD

Corresponding Author

POINT-BY-POINT RESPONSE TO REVIEWERS’ COMMENTS

Reviewer comments – AGYW Paper

1. Reviewer #1

Overall, the paper is well written. However, the authors examine a subject that is exhausted. It is not clear what is novel about this paper as the main findings from this paper have been well established in several papers as the authors rightly admit. Instead one might expect that interventions to keep AGYW in school and economic empowerment programs are a more critical topic and such interventions should be the subject of current inquiry. Also, there are some major data analytical flaws that need to be addressed.

Response: We agree with the reviewer that our area of inquiry has been explored before and that the main findings mimic those from previous studies. However, as noted in our paper, most previous studies did not include the very young ones (10-14 years) and were largely focused on one disease (either HIV or syphilis) or one population (in-school or out-of-school AGYW) but not both. Thus, our paper extends previous research by examining sexual-risk behaviors and the prevalence of HIV and syphilis while taking into consideration the above-mentioned gaps in previous studies. Our intention was to write a purely descriptive paper that program implementers would use and cite as they design target-specific interventions for AGYW. It is important to note that the data come from a study that was implemented to inform the design of interventions for AGYW in Uganda, funded through the Global Fund. So, yes, we will follow through with papers that include multi-level regression analyses but our intention initially was to present this paper as a purely descriptive one. 

Major comments

The authors present a standard definition of the AGYW as those aged 15 to 24 in the introduction. However, the authors enrolled participants from 10 years. Is it correct to refer to the study group with a wider age bracket as AGYW?

Response: The reference to 15-24 years in the ‘introduction’ section of our paper is because the literature is vast with studies on AGYW aged 15-24 years. Most program implementers usually lack data on a wide range of sexual and reproductive health behaviour of the very young girls aged 10-14 years, and this affects programming for these girls. Our paper tries to provide some of these data. Besides, the standard definition of ‘adolescents’ is 10-19 years, while that of ‘young people’ is usually 15-24 years; so, we believe that referring to our population as adolescent girls and young women aged 10-24 years is appropriate and we have kept it that way throughout the paper. 

In line 120, the authors state that further research is needed to understand the reasons for the differences in the risk of HIV and other STIs between in and out of school AGYW. This would have been a good addition to the body of knowledge, however it was not explored in this paper. To answer this question, one might have expected a qualitative inquiry.

Response: We have edited the statement to read, ‘…improve our understanding of the differences in risk-taking behaviors and the prevalence of HIV and other sexually transmitted infections (STI) between in- and out-of-school AGYW’ (see lines 112-113, page 4). We believe that since most studies include one (in-school or out-of-school) population, it is not always easy to examine the differences in risk-behaviors and the prevalence of HIV and other STIs among in- and out-of-school AGYW. This is what we intended to emphasize, and we hope the revised sentence helps to improve clarification.

Line 132-137 is a description of the strength of the study and should be moved to the discussion section. It is very unusual to present the number of study participants enrolled in the introduction section.

Response: The description in lines 132-137 has been deleted from the introduction section. Part of the description has been taken to the discussion section as recommended (see lines 595-595, page 28).

In line 188, the authors state for a total of “80 schools in 20 districts”. Was this planned, and if so, it should be stated explicitly. Also placing this information in the brackets takes away its significance.

Response: We have revised the statement to clarify that it was our intention to survey 80 schools in 20 districts. See lines 182-185, page 7.

In line 220, the authors explain the sampling at household level, however, it is not clear how the representation was achieved without using a stratified approach in their sampling approach.

Response: We have improved clarity in the sampling of households as needed. See lines 212-216, page 8.

In line 241, how did the investigators determine which AGYW would be interviewed by men, and which ones by females?

Response: We assigned female interviewers to the very young ones (10-14 years) and those aged 15-17 years while the male interviewers interviewed the older adolescents (18-19) and young women (20-24 years). This was done to ensure that the very young girls felt comfortable to respond to the questions. We did not experience any challenges with this arrangement.

The authors should provide examples of questions that the AGYW answered. It is not clear to the reader how these questions were structured especially to fit a young audience of 10 years, and if they really understood these questions.

Response: Where clarity was needed, we included an example, or a description of what the question asked for. We have provided an example of such questions in the revised paper (see lines 233-235, page 9). Detailed questions are included in the survey questionnaire which is submitted as a supplementary material as part of the original submission. Besides, the questions were translated and administered in the local language, which improved the girls’ understanding of the questions. These aspects have been clarified in the revised paper.

Ethical-legal issues arise from cases where minors report sexual abuse and it is not clear what was done. A statement on this issue is important.

Response: Where cases of sexual abuse or intimate partner violence were reported, we referred the girls to the nearest health facilities for management. A statement to this effect has been added at the end of ‘ethical considerations’.

Definition of study outcome: The sexual risk behavior study outcome appear to be several. In the analysis, all these items have remained separate. The authors did not attempt to create a composite index which would bring all these together.

Response: We intended to analyse for each sexual-risk behaviour separately given that each behaviour is sufficient to result in HIV/STI infection or teenage or unwanted pregnancy. Besides, such individual risk-behavior data can easily inform program implementers where they should focus most. We have improved our definition of what constituted a sexual-risk behaviour; so, we hope this will help to clarify why we did not create a composite variable that brings all sexual-risk behaviors together.

Sample size calculation: The data are clustered at multilevels namely district, schools or villages. This was not taken into account at the sample size calculation to adjust for the potential design effect.

Response: We have revised the ‘sample size estimation’ sub-section to indicate what we did to account for clustering at the district level. However, we did not account for clustering at school or village level, and this is likely to have affected the precision of our sample size estimation. We have acknowledged this as a limitation. See lines 585-592, page 27.

Data analysis: The authors have presented only crude results which could potentially be affected by confounding. One would expect the measured of effect would be attenuated if an adjusted analysis were conducted.

Response: As described above, our intention was to present descriptive statistics but not to conduct multivariable regression analyses. We acknowledge that this would have been essential to identify the factors independently associated with HIV or syphilis infection or the factors associated with engagement in sexual-risk behaviours. We have provided further explanation of why we opted for a descriptive study in lines 585-592, page 27.

In the same line with adjusting for confounding, the authors did not adjust for the clustering effect. This may be accomplished using the Generalized estimating equation (GEE) models

Also, the authors did not take into account the survey design especially if they used the enumeration areas. The “svy” option in STATA would help to offset that.

Response: As already explained, we acknowledge the importance of adjusting for confounding and the need to use GEE or any other models to adjust for clustering effect. However, the general purpose of the paper was to present descriptive statistics; so, we did not use any regression models for this reason. We have indicated that we used the ‘svy’ option when accounting for clustering at the district level but our study was not powered to do the same at school or village level, and this could have affected the precision of our sample size estimation. We have acknowledged this as a limitation. See lines 585-592, page 27, for details.

The % of HIV positives in-school were lower than those out-of-school. This may be explained by the higher risk sexual behaviors. However, is it also possible that HIV positive girls drop out of school when they learn about their HIV status. It is not clear here what the chicken and what is the egg. However, the authors seem to assert that girls drop out of school and then they become HIV infected.

Response: Since this was a cross-sectional study, we can’t tell if the girls were already infected before they dropped out of school or got infected because they dropped out of school. Although we did not see the assertion referred to by the reviewer in the original paper, we have ensured that the revised paper does not carry this assertion either.

The results within Table 2 and 3 have varying denominators depending on whether an AGYW has had sexual intercourse before. To avoid confusion, the authors should revise the table and include the (n) against the variable to clarify on how many answered this question since the denominator at the top of the table of schooling status is not applicable. For example the question age at first sex is only answered by those who had ever had sex, yet based on the table it appears as if the denominator is all the 8236, but is actually half of that.

Response: We have revised the presentation of data in the tables and also in the text to clarify which category of participants is being referred to. For instance, where the percentages refer to those that have ever had sex, or those that had ever had sex who reported engaging in sex in the past twelve months, this has been clarified. Please see Table 2 (page 17) and Table 3 (page 20) and their accompanying text for details.

The mode of presentation of results needs to take into consideration the denominator even in the text. For instance, in line 417 to 419 the authors state that “Nearly eighty-six per cent (n=3,848) of the AGYW reported that they were willing or somewhat willing to have sex at their sexual debut; with comparable proportions of out-of school and in-school AGYW (84.4%, n=1,245 vs. 86.3%, n=2,603)” There is a need to clarify that this is ‘among those who had ever had sex’. The same applies to line 431, and the authors should clarify this throughout the manuscript.

Response: As noted above, we have revised the presentation of the data in the tables, and we have revised the text accompanying each table. We hope this helps to clarify the numbers. See Table 2, page 17, and Table 3, page 20, and the accompanying text as mentioned above.

For table 4, some results appear in the text but not in the table. The RR’s should be presented in the table as well and not just the text. Also, please explain the rationale for RRs in the data analysis section, given this is a cross sectional study. What form of regression was used to generate these results and explain the choice. Why did the authors not conduct a multivariable regression analysis?

Response: We have dropped all reference to RR and retained only the p-values computed using Chi-square tests. We did not perform any regression analyses for reasons already described above.

As already mentioned, no multivariable results are presented. For instance, is wealth tertile independently associated with HIV infection regardless of the schooling status? The same would apply to the sexual risk behaviors. Overall, the data analysis lacks sufficient rigor and needs to be re-examined extensively.

Response: We agree with the reviewer that multivariable regression analyses would have helped to identify the factors independently associated with HIV and syphilis infection among in- and out-of-school AGYW. However, as already explained, this was not the purpose of this analysis. Our analysis was informed by the need to generate descriptive statistics necessary to inform the design of target-specific risk-reduction interventions for AGYW, stratified by schooling status. Otherwise, the reviewer’s message on the need to conduct regression analyses (and other suggestions about the design effect and the need to adjust for clustering, etc.) has been clearly noted. This will help to inform our next series of papers from the same dataset.

The argument in line 554-556 is inadequately presented and could as well be removed.

Response: The text in lines 554-556 has been dropped from the revised paper, as advised.

Minor comments

The second sentence (line 57/58) in the results in abstract section specifically refers to the out of school adolescents who are not able to read. It is not clear why the authors specifically focus on this subgroup. The reader might expect overall rate of literacy or present the two subgroups for comparison.

Response: All reference to AGYW’s readability was dropped from the abstract during the revision process.

2. Reviewer #2

General comments:

Thank you for the opportunity to review this piece of work. Overall, the paper is well-written and presents an important public health problem. However, it is very descriptive and could have been strengthened by a theoretical framework or be hypothesis-driven question and simple regression analysis to control for confounders.

Response: We thank the reviewer for these observations. We agree that use of a theoretical framework or a hypothesis-driven question could have helped to strengthen the analysis. Given that this paper was purely a descriptive paper, our analysis was not guided by any hypothesis-driven questions. We have acknowledged this as a limitation. Also, we did not do any regression analyses since the primary purpose of the analysis was to present descriptive statistics (stratified by schooling status) to inform the design of target-specific interventions. See lines 585-592 for details on the limitations of the study.

Specific comments:

1. Abstract – line 57 – it is not clear what the sample size is and how many AGYW were recruited in the study across the 20 districts? What is the expected age-range for in-school as only 50% in school sounds very low? Please clarify.

Response: We have revised the abstract to improve clarity on the numbers, so, some of the issues raised by the reviewer no longer exist in the paper. We have clarified that the analysis was done among 8,236 in- and out-of-school AGYW aged 10-24 years in 20 districts. This is the number of AGYW in the database. The sample size for the main study from which the data are drawn was 8,473 but the analysis is based on 8,236 who participated in the study (based on records in the AGYW database). See lines 164 (page 6) and 369 (page 14) for details on the two numbers (8,473 and 8,236).

During the revision process, the reference to 50% being in-school has been dropped but here is the clarification on this percentage: it wasn’t meant to be a percentage of all in-school AGYW in the districts surveyed. No. It was meant to show that 50% of the 8,236 AGYW interviewed were in-school AGYW. In summary, it was meant to show that equal proportions of in- and out-of-school AGYW were included in the survey. However, as noted, the reference to 50% in-school has been dropped during the revision process.

2. Abstract line 57/58 – was not being able to read text in local language a measure of literacy?

Response: We have dropped all reference to literacy in the abstract. However, we have clarified in the main text that this was meant to serve as a proxy for literacy (see line 375, page 15).

3. Line 165 – definition of out-of-school is it only based on duration out of school? How do you account for those who had completed the highest level and hence did not need to be in school? Maybe good to distinguish between out of school out of employment and completed school with dropouts. Adding an age factor may assist in defining who is a school drop out and hence more vulnerable maybe, and also identify repeat graders – those staying in school beyond the age-grade level. Do you have data on those who were still in school rather than ‘highest level education reached’?

Response: We have clarified on the definition of out-of-school AGYW as those that had not completed school but who had been out of school for at least one year. In other words, our interest was in those that would have been in school at the time but were out of school; not those that had completed school. See lines 153-156, page 6, for details.

4. Line 245 editing questionnaires or maybe quality control? Please clarify

Response: We intended to refer to quality checks in the field: interviewers reviewed each other’s completed questionnaires to check for completeness and any questions that the interviewer might have inadvertently missed. However, the word ‘editing’ has been dropped to avoid confusing the readers as some of them may think that we were editing the questions in the questionnaire to improve clarity.

5. Line 366 – I am not sure what we are measuring with ability to read in local language – is it literacy – if so please indicate? What was done with those who couldn’t read/write when obtaining consent?

Response: As explained above, we used a girl’s ability to read in their local language as a proxy measure of literacy (see line 375, page 15), and this has been clarified in the main text. For consenting process, we had a provision for use of a thumbprint for illiterate girls (i.e. the informed consent form was read to the illiterate girls and they were asked to thumbprint as a sign of expressing their consent to participate in the study).

6. Line 432 – do we know the relationship with the most recent partner as it determines whether condoms are used or not and to what extent they are used?

Response: We have included data on the relationship with the most recent partner as requested. See lines 434-437, page 19.

7. I have a problem with comparing HIV and STI prevalence (and any key outcomes) between in and out of school without controlling for key confounders such as age, important sexual behaviors that predispose AGYW to HIV infection as well structural factors that make AGYW vulnerable. A simple multivariable analysis would have taken the analysis and interpretations a step further.

Response: We appreciate the reviewer’s comment on the lack of multivariable analysis. However, as described earlier, our intention was to present descriptive statistics, as our aim was not to identify factors independently associated with HIV or syphilis infection among AGYW. For this reason, no regression analyses were conducted. We have acknowledged this as a limitation (see lines 585-592, page 27, for details).

8. In discussion – the authors refer to first-time protected sex, does this have any implication in terms of risk of acquiring HIV or syphilis? I think the message to drive is the need for correct and consistent use of protection and differentiated care for AGYW in and out of school.

Response: We have discussed the implications of having protected sex at all times, and the need to emphasize correct and consistent condom use among in- and out-of-school. See lines 525-528, page 25.

9. Line 536-540 since this study was cross-sectional it is important to be cautious of making assumption that HIV prevalence was high due to factors that were not measured in time in the study.

Response: The text in lines 536-540 has now been dropped from the revised paper.

3. Reviewer 3 

This is a cross-sectional study in which the authors sought to compare and link sexual risk behaviours in school going and out-of-school adolescent girls and young women in Uganda with HIV and syphilis biomarkers. The study strengths lie in the large sample size and the wide age group range that caters for different subgroups. The study is important as it seeks to encourage target specific interventions against sexual risk behaviours. The authors reported differences in sexual risk behaviours and HIV and syphilis between the different study groups but did not include enough statistical data to indicate if the differences were significant. I recommend the following revisions and clarifications:

Response: We thank the reviewer for this observation. We admit that we did not conduct any multivariable regression analyses, since this was not the primary purpose of the analysis, but we compared proportions on sexual-risk behaviors of interest using Chi-square tests. Our purpose in this analysis was to present descriptive statistics on the sexual-risk behaviors and HIV and syphilis prevalence to inform the design of appropriate HIV/STI interventions. However, we have acknowledged the lack of multivariable regression analysis as a limitation. See lines 585-592, page 27, for details.

Specific recommendations

1. Abstract - Include introduction statement/background information before getting to the study aim.

Response: We have included a background/introduction statement in the abstract as advised. This is placed before the study aim is stated.

2. Some Risk ratios in abstract results do not appear in the main results section

Response: We have dropped all reference to risk ratios since this is a descriptive study. Instead, we have reported p-values to show the level of statistical significance between any two groups. The p-values were derived using Chi-square tests.

3. Questionnaire validation - How many and how were participants selected for field pilot studies. Please indicate

Response: We have included additional information on the number of AGYW that were interviewed as part of the piloting of the study tool. These participants were contacted via the health teams in the community selected for pilot-testing. See lines 235-238, page 9.

4. Line 245 indicates that editing of questionnaires was done in the field. Wasn’t the questionnaire finalized before beginning of field work and how were differences in the questionnaires between field teams handled?

Response: We have dropped all reference to the editing of the questionnaires. What we did was a quality control check to ensure that all questionnaires were complete. To avoid confusion, we have removed the language referring to ‘editing’ of questionnaires as it might be interpreted to refer to editing of questions to improve clarity. What we did was just a quality check.

5. Line 309 – For the statement, “rejecting the two most common misconceptions about HIV transmission or prevention”. What were these misconceptions? Please indicate

Response: We have added the two most common misconceptions, as advised. See lines 309-310, page 12.

6. Line 457 – HIV and wealth tertile, Please indicate if this difference was significant

Response: We have used Chi-square tests to determine if the reported HIV prevalence by schooling status differed across the different wealth tertiles. We have found that HIV prevalence was significantly lower among in-school than out-of-school across the different tertiles. See lines 475-481, page 21 for details.

7. Table 4 heading – Please revise heading by additionally including link to sexual risk behaviours

Response: We have edited the title of Table 4 as advised. It now reads as: ‘Distribution of HIV and syphilis prevalence by schooling status and selected sexual-risk behaviors’. See Table 4, page 23 for details.

8. Table 4 – Please indicate if reported differences were significant

Response: Because of multiple categories that include the disease (HIV or syphilis), schooling status and selected sexual-risk behaviour characteristics, we were unable to include a p-value for each value reported in Table 4. Instead, we opted to compare selected proportions, and where this comparison was made, we reported a p-value that indicates if the two proportions compared were statistically significantly different. 

9. Discussion – Line 477 – 487 and 580 - 584. It’s important for these findings and conclusions to be supported by statistical analysis indicating if differences between the groups were significant. This is especially so since the major aim of the study was to link sexual risk behaviours with biomarkers in terms of HIV and syphilis prevalence 

Response: The main findings, as summarized in lines 477-487 (opening paragraph of the discussion section) and also included in the conclusions (line 580-584) are based on statistically significant differences between in- and out-of-school AGYW. As we noted earlier, we performed statistical comparisons between the respective groups reported using Chi-square tests, and the p-values reported in the ‘results’ section are a result of these statistical comparisons.

Minor comments

1. Remove duplicated words, “to reduce” in the abstract conclusion - line 69

Response: The abstract has been revised extensively. The statement in which the phrase ‘to reduce’ had been duplicated was deleted during the revision process.

---

## [Decision Letter · Decision Letter 1]

31 Aug 2021

Sexual-risk behaviours and HIV and syphilis prevalence among in- and out-of-school adolescent girls and young women in Uganda: a cross-sectional study

PONE-D-21-03834R1

Dear Dr. Matovu,

We’re pleased to inform you that your manuscript has been judged scientifically suitable for publication and will be formally accepted for publication once it meets all outstanding technical requirements.

Kind regards,

Susan Marie Graham, MD, MPH, PhD

Academic Editor

PLOS ONE

Additional Editor Comments (optional):

Reviewers' comments:

Reviewer's Responses to Questions

**Comments to the Author**

1. If the authors have adequately addressed your comments raised in a previous round of review and you feel that this manuscript is now acceptable for publication, you may indicate that here to bypass the “Comments to the Author” section, enter your conflict of interest statement in the “Confidential to Editor” section, and submit your "Accept" recommendation.

Reviewer #1: All comments have been addressed

Reviewer #3: All comments have been addressed

2. Is the manuscript technically sound, and do the data support the conclusions?

Reviewer #1: No

Reviewer #3: (No Response)

3. Has the statistical analysis been performed appropriately and rigorously? 

Reviewer #1: No

Reviewer #3: (No Response)

4. Have the authors made all data underlying the findings in their manuscript fully available?

Reviewer #1: Yes

Reviewer #3: (No Response)

5. Is the manuscript presented in an intelligible fashion and written in standard English?

Reviewer #1: Yes

Reviewer #3: (No Response)

6. Review Comments to the Author

Reviewer #1: I have read the revised version. There is concern that the authors do not address the major concern that the results are not adjust for obvious confounding from several variables. The results presented do not accurately reflect the associations described. For instance, it is not clear whether out of school AGYW will carry higher chances of having STDs independent of socio-economic status.

One expects that multivariable regression is standard statistical practice especially with a large sample such as that in this study.

Reviewer #3: (No Response)

7. PLOS authors have the option to publish the peer review history of their article (what does this mean?). If published, this will include your full peer review and any attached files.

Reviewer #1: No

Reviewer #3: No

---

## [Editor Report · Acceptance letter]

3 Sep 2021

PONE-D-21-03834R1 

Sexual-risk behaviours and HIV and syphilis prevalence among in- and out-of-school adolescent girls and young women in Uganda: a cross-sectional study 

Dear Dr. Matovu:

I'm pleased to inform you that your manuscript has been deemed suitable for publication in PLOS ONE. Congratulations! Your manuscript is now with our production department. 

Kind regards, 

on behalf of

Dr. Susan Marie Graham 

Academic Editor

PLOS ONE